# OpenReview forum: "Fine-grained Analysis of Brain-LLM Alignment through Input Attribution"
_ICML.cc/2026/Conference — ICML 2026 regular_

### Official Review · Reviewer_KAGv · 2026-03-07

**Soundness:** 3
**Presentation:** 4
**Significance:** 3
**Originality:** 3
**Overall Recommendation:** 5
**Confidence:** 3

**Summary:**

This study deploys gradient attributions to determine which words in the input sequence influence (i) a model’s next word predictions, and (ii) a model’s representational alignment with brain activity. Using a series of 1-2B-parameter Transformers and SSMs, it is found that the words important for both tasks are largely disjoint, suggesting that brain alignments are based on more than just structural information. Further analyses provide evidence that syntax-, semantic-, and discourse-relevant words are more important for brain alignments, but mostly syntax-relevant words are important for next-word prediction. Overall, this study sheds light on the the kinds of features that influence recent findings related to brain-LLM representational alignments.

**Compliance With Llm Reviewing Policy:**

Affirmed.

**Final Justification:**

The new evidence and clarifications provided during the rebuttal addressed some minor concerns; my overall evaluation remains the same (largely positive).

**Key Questions For Authors:**

Why were 4 TR embeddings used, rather than some other number?

**Limitations:**

Limitations and societal impacts are directly discussed in the paper. I found these sufficient.

**Strengths And Weaknesses:**

Strengths:
===
* I have reviewed quite a few LLM-brain alignment papers, and this is the first that I have found genuinely interesting and enjoyable to read the whole way through.
* As someone who does not work directly in the neuroscience/LLM intersection, I found the motivation, related work, and contributions clear; I think this implies that a wide audience will find this study approachable.
* Relatedly, the writing is clear throughout, with only a few key exceptions detailed below.
* As far as I am aware, the approach for making gradient attributions computable for LLM-brain alignments (using a linear projection to map from LLM activations to voxel activity, then computing the MSE) is novel. It’s not a huge point of novelty, but it’s sensible and simple.
* It is interesting that brain alignments cannot be reduced merely to next-word prediction; this suggests that many of the recent papers on this topic have been uncovering non-trivial representational alignments.

Weaknesses:
===
* Technical novelty is limited. The methods used are appropriate for investigating the research questions, so I do not think this is a significant concern.
* It would be nice to see more diversity in the representational power of the models tested. Even with the stated computational limitations, one could still analyze models smaller than 1-2B parameters. One could also reduce precision or quantize models such that analyzing larger models becomes feasible.
* There is no reason not to prefer integrated gradients over gradients x inputs. Gradients x inputs has known flaws that integrated gradients addresses with very limited additional computation. I see in the appendix that runtime sometimes differed significantly (though not always in the same direction, strangely), but this can be addressed by reducing the number of iterations; even 5 to 10 is usually sufficient to see significant improvements. I appreciate that the authors did include this method; what I am suggesting is that these become the main results, and that gradients x inputs either be demoted to the appendix or removed entirely.

Comments and Suggestions:
===
* To summarize trends across models and thresholds in Fig. 2, consider reporting the area under the IoU curve for each model, and testing the significance of their differences to the baseline and each other.
* Consider explaining in the main paper how words are sorted into syntactic, semantic, and discourse-related categories. I eventually found a reference to an appendix where this was explained, but readers might appreciate if this information were emphasized.

---

> ### Author Rebuttal · Authors · 2026-03-30
>
> _We thank the reviewer for the useful suggestions and constructive feedback._
>
> # Models
> During the review period, we extended the analysis to additional Gemma variants (2B-Instruct and 7B) and observed the same qualitative pattern: low BA-NWP overlap at stringent attribution thresholds, higher attribution spread for NWP at early layers and for BA at middle/late layers, stronger BA reliance on semantic/discourse features, and a broader BA recency profile. These initial results suggest that the reported BA-NWP differences may extend beyond the 1-2B scale and to other training objectives. Full details are provided in the response to reviewer qxCS.
>
> # Paper organization and presentation
> In the revised manuscript, we will make the following changes:
> - Move results with Integrated Gradients to the main paper in place of Gradient X Input, which will be moved to the Appendix.
> - Add the area under the IoU curve for each model, and test the significance of their differences to the baseline and each other.
> - Move the categorization into syntactic, semantic, and discourse-related categories to the main paper.
>
> # Why use 4 TR embeddings?
> We thank the reviewer for this question. The use of 4 TRs follows standard practice in fMRI encoding models of language and is motivated by the hemodynamic response delay. Because the BOLD signal reflects neural activity with a latency of several seconds, concatenating embeddings from multiple preceding TRs allows the encoding model to capture the temporal spread of the response. In particular, using 4 TRs (≈8 seconds in the Harry Potter and Moth Radio Hour datasets) provides a good empirical approximation of the peak and early tail of the hemodynamic response and has been widely adopted in prior brain–language alignment work [1,2].
>
> [1] Toneva, M., & Wehbe, L. (2019). Interpreting and improving natural-language processing (in machines) with natural language-processing (in the brain). Advances in neural information processing systems, 32.
>
> [2] Aw, K. L., Montariol, S., AlKhamissi, B., Schrimpf, M., & Bosselut, A. Instruction-tuning Aligns LLMs to the Human Brain. In First Conference on Language Modeling.

---

> > ### Author Rebuttal · Reviewer_KAGv · 2026-04-03
> >
> > Thanks for the response. 7B models are within the same order of magnitude of parameters, so I don't know that this additional evidence fully addresses that (minor) concern.
> >
> > I appreciate the promise to address clarity-related comments, and the clarification on why 4 TR embeddings were used.
> >
> > My original review was quite positive, and I think the scope of contributions in this paper and the discussion warrants a similar score as I have given.

---

> > > ### Author Response · Authors · 2026-04-05
> > >
> > > We thank the reviewer for the positive assessment and helpful feedback.
> > >
> > > Regarding model scale, we agree that extending the analysis to substantially smaller and larger models would further strengthen the study. Due to the computational cost of the attribution pipeline and the limited rebuttal time window, we prioritized additional experiments aimed at addressing the reviewers’ main concerns, including extending the analysis to Gemma-2B-Instruct and Gemma-7B (more diverse, bigger models). We will explicitly include broader scaling analyses as future work in the Discussion section of the revised manuscript. We thank the reviewer for their understanding regarding these practical constraints.
> > >
> > > As a side note, we uploaded an updated version of Figure 2 with the suggested modifications at [this link](https://anonymous.4open.science/r/new_results-4819/README.md), under _new\_models\_/all\_models\_positional\_analysis.png_.

---

### Official Review · Reviewer_CwJs · 2026-03-11

**Soundness:** 2
**Presentation:** 3
**Significance:** 3
**Originality:** 2
**Overall Recommendation:** 4
**Confidence:** 4

**Summary:**

The paper proposes a pipeline to compare the word-level input depedence of brain alignment (BA) and next token prediction (NTP). By comparing the layerwise attribution scores on the Harry Potter dataset to language modeling and fMRI voxel activation prediction measured by the IoU and CoM metrics, the authors show that BA and NTP employs different sets of words in terms of linguistic roles and positions.

**Compliance With Llm Reviewing Policy:**

Affirmed.

**Final Justification:**

The authors' rebuttal solves some of my concerns, and I think the soundness of the paper is strengthened.

**Key Questions For Authors:**

- Since the tested models are not optimized for the BA task, their BA behavior can be regarded as a 'by-product' of NTP training. As a result, how do you explain the mechanism with which the models develop a different context perception pattern from NTP?
- Figure 26(b) shows the BA correlations of the five models, which are all above 1. Does it mean that all these models outperforms the noise ceiling (human-human similarity)?

**Limitations:**

Yes

**Strengths And Weaknesses:**

## Strengths
- The paper raises an interesting question about the difference between the input reliance patterns of BA and NTP in LLMs. The findings about linguistic roles and positions gives novel insights about model-brain similarity.
- The two proposed metrics are suitable for their purposes.
- The experiments cover multiple LLMs.

## Weaknesses
- Interpretation of the experiment results relies on the atrribution threshold. However, the paper does not propose a preferred or reference threshold level, which makes some of the findings ambiguous. For example, Figure 2 is supposed to show the low overlap between BA and NTP. However, when the atrribution threshold becomes 80%-100%, the overlap becomes 60%-80%, which cannot be simply described as 'low'.
- The attribution of BA, which includes concatenating embeddings, ridge regression and linear projection, is quite different from the attribution of NTP, which could naturally bring difference to the results, which the paper does not take into account.
- The experiments are performed on limited text stimuli and human subjects.

---

> ### Author Rebuttal · Authors · 2026-03-30
>
> _We thank the reviewer for the useful comments and constructive feedback._
>
> # Attribution thresholds
> Our goal in Fig. 2 was not to select a preferred attribution threshold, but to analyze overlap across the full range of cumulative attribution mass. For this reason, we report IoU curves across thresholds rather than fixing one reference value.
>
> Our claim of low overlap refers primarily to stringent attribution regimes (top 1–10%), which identify the smallest subset of words carrying the most attribution mass. In this range, IoU remains around 0.1–0.2, indicating that BA and NWP rely on largely distinct high-impact words. At higher thresholds (80–100%), overlap increases naturally because these regimes include most of the context.
>
> We will clarify this distinction between high-impact-token overlap and broad-context overlap in the revised manuscript.
>
> # Relation between BA/NWP attribution pipelines and attribution differences
> We agree that the pipeline asymmetry between BA and NWP, inherent to brain encoding constraints, may contribute to the observed effects. However, converging evidence suggests it does not drive the main findings, as discussed below. We will discuss this aspect in Section 3.3 and Limitations of the revised manuscript.
>
> 1. If four-TR temporal smoothing were the primary driver of BA attribution spread, we would expect a uniform increase across layers and a shift toward the most frequent feature class (syntactic, 95%). Instead, attribution spread reverses across layers (higher for NWP early, higher for BA at later layers), and BA shows selective increases in less frequent semantic (60%) and discourse (78%) features, inconsistent with a smoothing artifact.
>
> 2. Both pipelines are fully differentiable end-to-end, so attribution reflects each task's sensitivity to input words with respect to its final objective. The ridge penalty in the BA pipeline shrinks output weights uniformly toward zero, attenuating gradient magnitudes globally but not selectively across input positions or linguistic categories, making it an unlikely driver of the systematic positional and linguistic differences we report.
>
> # Brain datasets
> We agree that evaluating additional datasets and larger subject pools would further strengthen the generality of the results. However, this limitation reflects current dataset availability rather than a design choice.
>
> There are currently no public fMRI datasets with substantially longer naturalistic stimuli suitable for word-level alignment analyses. The Harry Potter (8 subjects) and Moth Radio Hour (9 subjects) datasets are widely used benchmarks in brain–LLM alignment research (e.g., [1]). Importantly, the two datasets differ substantially in stimulus type (fictional narrative vs. spoken autobiographical stories), yet we observe consistent BA–NWP attribution differences across both, suggesting the results are not tied to a single stimulus or participant group. Our framework can be applied directly as more diverse datasets become available.
>
> [1] Zhou Y. et al. Divergences between language models and human brains. NeurIPS 2024.
>
> # Why BA differs from NWP if models were trained only on NWP
> Although the tested models are trained only for NWP, our results suggest that BA and NWP rely on overlapping representations but different subsets of contextual evidence extracted from those representations.
>
> NWP training encourages models to encode multiple types of linguistic information simultaneously (syntactic, semantic, and discourse-level), even though only a subset of this information is directly required for predicting the next token at inference time. The brain-encoding model can then selectively read out the components of these representations that better match neural responses. In this sense, BA does not reflect a separate mechanism learned by the model, but rather a different functional use of shared representations.
>
> Consistent with this interpretation, BA shows broader attribution spread and stronger reliance on semantic and discourse context than NWP. This suggests that brain alignment emerges from higher-level contextual structure already present in pretrained representations, rather than from the predictive objective alone. We have clarified this interpretation in the revised manuscript.
>
> # BA correlations exceeding noise ceiling
> Figure 26(b) reports correlations normalized by the noise ceiling. Values above 1 indicate that unnormalized correlations slightly exceed the estimated ceiling, which can occur because the ceiling is itself a statistical estimate derived from finite noisy data rather than a strict upper bound. Similar effects have been reported in prior work [2], highlighting the need for improved ceiling estimation and larger neural datasets.
>
> [2] Aw K. L. et al. Instruction-tuning Aligns LLMs to the Human Brain. In COLM 2024.

---

> > ### Author Rebuttal · Reviewer_CwJs · 2026-04-02
> >
> > The rebuttal solves some of my concerns, but I still have some questions:
> >
> > 1. Brain datasets: Although some commonly used fMRI datasets have small samples sizes, there *are* some larger datasets, for example, the "Narratives" dataset[1,2]. It will be always better to validate a hypothesis with a larger sample size given proper resources.
> >
> > 2. The interpretation "brain alignment emerges from higher-level contextual structure already present in pretrained representations, rather than from the predictive objective alone" seems to indicate that NTP (which actually includes comprehension of syntactics and semantics) is a secondary capability obtained during the pretraining process, and BA is closer to a higher-level capability, which is interesting. Could you give some discussion on what the higher-level capability could be and how to measure them?
> >
> > 3. A brain-correlation score higher than the noise ceiling is rather rare if the score is calculated with train-test data split of the regression model, e.g., in [2] they report scores around 60% of the noise ceiling. Does this paper uses train-test data split in the regression?
> >
> > [1] Nastase, S.A., Liu, YF., Hillman, H. et al. The “Narratives” fMRI dataset for evaluating models of naturalistic language comprehension. Sci Data 8, 250 (2021). https://doi.org/10.1038/s41597-021-01033-3
> >
> > [2] Caucheteux, C., Gramfort, A. & King, JR. Evidence of a predictive coding hierarchy in the human brain listening to speech. Nat Hum Behav 7, 430–441 (2023). https://doi.org/10.1038/s41562-022-01516-2

---

> > > ### Author Response · Authors · 2026-04-03
> > >
> > > _We thank the reviewer for the constructive discussion and thoughtful follow-up questions. Below we clarify the three remaining points._
> > >
> > > ### 1.
> > > We totally agree with the reviewer’s point, and will more explicitly highlight the need to test on larger subject pools in the Limitations. It would be particularly interesting for future work to use our attribution pipeline with the “Narratives” dataset to test whether the same results also hold with different stimulus modality (reading vs listening) and expanding the analysis to audio models. In our work, we tried to find an experimental setup that allowed us to test multiple models, datasets, and attribution methods within the limits of the available resources.
> > >
> > > ### 2.
> > > Our attribution analyses show that BA depends more strongly on broader semantic and discourse-level context than NWP. Since such information is already encoded in pretrained representations learned through the NWP objective, we interpret BA as reflecting a different functional readout of shared representations.
> > >
> > > One way to operationalize and measure these “higher-level capabilities” is to examine how different downstream tasks read out the same underlying representations. For example, starting from the NWP-pretrained feature extractor, we can train task-specific heads on diverse downstream tasks (e.g., summarization, question answering) and compute attributions with respect to the new objectives. Our hypothesis is that tasks requiring global semantic and discourse integration will exhibit attribution patterns more similar to BA than to standard NWP.
> > >
> > > Additionally, prior work has shown that fine-tuning on summarization improves brain alignment (Aw & Toneva, 2023 [ICLR]). A natural extension would be to analyze whether such improvements correspond to systematic shifts in attribution patterns (e.g., broader contextual spread or increased reliance on long-range dependencies), providing further evidence that BA is linked to how higher-level contextual information is utilized rather than to the predictive objective itself.
> > >
> > > We will clarify this interpretation and include this discussion in the revised manuscript.
> > >
> > >
> > > ### 3.
> > > Yes, the encoding models use train–test splits with nested cross-validation, following standard brain-encoding practice. As described in Section 3 of Aw et al. (2024), a linear regression model is fit from LLM layer activations to fMRI responses and then evaluated on held-out data to compute brain alignment as the Pearson correlation between predicted and measured activity.
> > >
> > > Noise-ceiling values also depend strongly on the dataset and subject variability, which makes direct comparison across studies difficult (e.g., Caucheteux et al. (2023) uses the Narratives dataset rather than Harry Potter). Moreover, the noise ceiling is itself an estimate derived from inter-subject correlations and therefore does not represent a strict upper bound. For example, Aw et al. (2024, App. G Table 5) show that several recent models (e.g., LLaMA, Vicuna) exceed the estimated noise ceiling on the Harry Potter dataset (Wehbe et al., 2014), while different models exceed the ceiling on other datasets (e.g., Blanck et al., 2014), illustrating that this effect varies across datasets and architectures.
> > >
> > > Finally, Caucheteux et al. (2023) evaluate earlier-generation language models (e.g., GPT-2, XLNet, Transformer-XL), whereas our study uses more recent architectures. Prior work (Aw et al., 2024) shows that models such as GPT-2 and even larger earlier architectures (e.g., T5-XXL) do not exceed the estimated noise ceiling on the Harry Potter dataset, while more recent models (e.g., LLaMA, Alpaca, Vicuna) do. This difference in model class likely contributes to variation in normalized scores across studies.
> > >
> > > _We hope these clarifications address the remaining questions and support a more positive assessment of the paper._

---

### Official Review · Reviewer_qxCS · 2026-03-12

**Soundness:** 2
**Presentation:** 2
**Significance:** 3
**Originality:** 2
**Overall Recommendation:** 4
**Confidence:** 3

**Summary:**

This paper studies brain-LLM alignment by comparing brain alignment (BA) and next-work prediction (NWP). The results show that BA prioritizes semantic and discourse-level information while NWP focuses more on syntax.

**Compliance With Llm Reviewing Policy:**

Affirmed.

**Final Justification:**

The author's rebuttal addresses my prior misunderstanding regarding the position of this paper. Therefore, I would like to increase my score.

**Key Questions For Authors:**

In input attribution, the author uses the embeddings of different layers. However, the attention will mix token representations after the first layer. How does the author correspond the contribution of the embedding back to the input tokens?

**Limitations:**

yes

**Strengths And Weaknesses:**

Strength:

The analysis is thorough, the methods are elaborated in details.

Weakness:

1. Although the analysis is thorough, the findings are largely unsurprising and appear to confirm intuitions or common-sense expectations rather than providing novel insights.
2. The analysis is primarily limited to the Harry Potter narrative. Expanding the study to other types of texts (e.g., scientific reports, political analysis, or news articles) would strengthen the contribution and allow for meaningful comparisons across genres.
3. The model sizes used in the experiments are relatively small. It would be beneficial to evaluate larger language models (e.g., models around 7B parameters) to verify whether the observed findings hold at larger scales or whether different behaviors emerge.
4. The author could consider including more discussions or preliminary experiments on how to leverage such findings to improve the performance/robustness/explainability of AI models.
5. There is a formatting issue in the references: “fMRI” should appear in uppercase. In BibTeX, this can be enforced by wrapping it in curly braces (e.g., {fMRI}).

---

> ### Author Rebuttal · Authors · 2026-03-30
>
> _We thank the reviewer for the thoughtful suggestions and careful feedback._
>
> # Novelty of findings
> While some qualitative trends may appear intuitive, our contribution is to demonstrate them quantitatively, systematically, and at word-level resolution, which has not previously been done in brain-LLM alignment research. Specifically:
>
> 1. While prior work compared objectives indirectly (e.g., via model–brain correlations), we show directly that the sets of words most important for BA and NWP overlap only weakly at stringent attribution thresholds.
>
> 2. We provide the first word-level comparison of positional biases in BA and NWP, revealing different contextual integration strategies.
>
> 3. We show that BA systematically allocates more attribution mass to meaning-oriented features, while NWP prioritizes syntactic features. Prior work suggested this possibility via perturbation-based strategies but did not test it directly at the word level.
>
> 4. We demonstrate cross-architecture consistency (transformers, SSMs, and hybrid models).
>
> Finally, our main methodological contribution is a pipeline to apply attribution methods to brain-LLM alignment that can be applied to a wide range of future questions beyond those addressed here.
>
> # Brain datasets
> We agree that extending the study to additional genres would strengthen generality. However, this limitation reflects current dataset availability rather than a design choice. Most public long-form fMRI language datasets use narratives because they maintain participant engagement during extended recordings, and they therefore constitute the standard benchmark in brain–LLM alignment work.
>
> Importantly, our analysis is not limited to a single stimulus: in addition to Harry Potter (fictional literary text presented in a controlled reading paradigm), we evaluate the Moth Radio Hour dataset, which contains spoken autobiographical stories with different discourse structure, lexical statistics, and narrative style. The consistency of BA-NWP attribution differences across both datasets suggests the results are not tied to a single text or style. We will clarify this limitation and highlight cross-genre evaluation as future work.
>
> # Models
> During the review period, we additionally evaluated Gemma-7B and Gemma-2B-Instruct, observing the same attribution patterns: area under the IoU curve in Fig. 2 significantly above the baseline (p<0.05); higher attribution spread for NWP at early layers and for BA at middle and late layers (early layers AUC: NWP $\approx 4500$, BA $\approx 2000$; late layers AUC: NWP $\approx 2300$, BA $\approx 3750$); BA draws more on semantic and discourse features than NWP (mean percentage of important words belonging to the two categories across the two models: NWP $\approx 25$%, BA $\approx 30$%), while NWP emphasize syntactic features (mean percentage of important words representing syntactic features: $\approx 40$% for NWP vs $\approx 24$% for BA); BA’s broader recency peak ($CoM=98\pm 146.7$) and NWP’s generally more pronounced primacy effect ($CoM=144.5 \pm 190.5$).
>
> In the revised manuscript, we will add this new initial evidence that BA-NWP differences might persist beyond the 1-2B scale and across training objectives.
>
> # Discussion on how to leverage findings to improve performance/robustness/interpretability of AI models
> We agree this is an important direction and will expand the discussion accordingly. Our results suggest three concrete avenues: (1) incorporating discourse-level or coherence-based supervision beyond next-word prediction objectives; (2) encouraging more adaptive context usage rather than edge-position reliance; (3) using brain-alignment attribution to identify whether models rely on linguistically meaningful evidence.
>
> # Clarification on the use of different layer embeddings
> We thank the reviewer for this important question. For both BA and NWP, token attributions are computed via end-to-end backpropagation from the task loss (MSE or cross-entropy) to the input token embeddings. Although BA uses representations extracted from intermediate layers, gradients propagate through the regression head and preceding LLM layers back to the input tokens, so attribution scores reflect each token’s contribution after contextual mixing by attention.
>
> # Formatting issue
> We thank the reviewer for spotting this issue. It has been corrected in the revised manuscript.

---

> > ### Author Rebuttal · Reviewer_qxCS · 2026-04-03
> >
> > I acknowledge the author's response to my concerns and questions. However, my concerns are not fully resolved.
> >
> > 1. **novelty of findings**:  The rebuttal reiterates the stated contributions but does not resolve my core concern. The main finding appears intuitive rather than novel. It is expected that next-token prediction emphasizes syntactic features, and this comparison does not convincingly inform the alignment between human brains and LLMs. A more appropriate comparison would align objectives across branches—for example, if the brain branch predicts neural activity, the LLM branch should analogously predict model-internal activity.
> >
> > 2. The rebuttal policy permits anonymous links for tables and figures. The authors should provide their new results via such links rather than describing them only in text.
> >
> > 3. I would expect at least preliminary experiments to support claims about improving performance, robustness, or interpretability, rather than relying solely on discussion to demonstrate the contributions of this paper.

---

> > > ### Author Response · Authors · 2026-04-05
> > >
> > > _We thank the reviewer for the follow-up questions. In the following, we address the remaining concerns._
> > >
> > > ## Novelty of findings
> > >
> > > We clarify that the central finding of this paper is not the observation that next-word prediction relies on syntactic information per se. Rather, our work introduces several novel results that inform the relationship between brain alignment and next-word prediction. This is also highlighted by other reviewers, e.g. “the findings about linguistic roles and positions gives novel insights about model-brain similarity” (CwJs), and “it is interesting that brain alignments cannot be reduced merely to next-word prediction; this suggests that many of the recent papers on this topic have been uncovering non-trivial representational alignments” (KAGv).
> > >
> > > Additionally, our framework to apply attribution methods to brain alignment represents a novel methodological contribution, and given its dataset- and architecture-agnostic nature it can be applied to investigate other questions in brain-LLM alignment interpretability.
> > >
> > > Finally, regarding the suggestion to compare neural activity prediction with model-internal activity prediction, we are not fully certain how this comparison would address the central research question of the paper. Our goal is specifically to analyze the relationship between brain alignment and next-word prediction, which is a broadly studied research question in the literature on brain-LLM alignment [e.g. 1-6], as discussed in the Introduction.
> > >
> > > ## Presenting new results in an anonymous repo
> > > Thanks for the suggestion. We now provide all new results for the two additional models at [this link](https://anonymous.4open.science/r/new_results-4819/README.md). We will update all plots in the main paper by including Gemma-2B-Instruct and Gemma-7B.
> > >
> > > ## Relation to improved AI models performance
> > > The example application of our attribution framework in our work is brain-LLM interpretability, with a focus on the relationship between brain alignment and NWP, which is a contentious research question in brain-LLM alignment research (e.g., 1-6).
> > >
> > > The research directions discussed in the rebuttal are complementary and require designing and testing new objective functions that allow integrating contextual representations in a more brain-like way. Therefore, they are beyond the scope of the current work and cannot be addressed within the rebuttal window.
> > >
> > > [1] Schrimpf, M., Blank, I. A., Tuckute, G., Kauf, C., Hosseini, E. A., Kanwisher, N., Tenenbaum, J. B., and Fedorenko, E. The neural architecture of language: Integrative modeling converges on predictive processing. Proceedings of the National Academy of Sciences, 118(45):e2105646118, 2021. doi: 10.1073/pnas.2105646118.
> > >
> > > [2] Goldstein, A., Zada, Z., Buchnik, E., Schain, M., Price, A., Aubrey, B., Nastase, S. A., Feder, A., Emanuel, D., Cohen, A., et al. Shared computational principles for language processing in humans and deep language models. Nature neuroscience, 25(3):369–380, 2022.
> > >
> > > [3] Hosseini, E. A., Schrimpf, M., Zhang, Y., Bowman, S., Zaslavsky, N., and Fedorenko, E. Artificial neural network language models predict human brain responses to language even after a developmentally realistic amount of training. Neurobiology of Language, 5(1):43–63, 2024.
> > >
> > > [4] Oota, S., Gupta, M., and Toneva, M. Joint processing of linguistic properties in brains and language models. NeurIPS 2024.
> > >
> > > [5] Merlin, G. and Toneva, M. Language models and brains align due to more than next-word prediction and word-level information. EMNLP 2024.
> > >
> > > [6] Zhou Y. et al. Divergences between language models and human brains. NeurIPS 2024.

---

### Official Review · Reviewer_dw3b · 2026-03-13

**Soundness:** 3
**Presentation:** 3
**Significance:** 3
**Originality:** 3
**Overall Recommendation:** 4
**Confidence:** 4

**Summary:**

This paper introduces an attribution pipeline for brain-LLM alignment (BA), aiming to identify which input words are most important for predicting brain activity from frozen LLM representations. As a case study, it compares BA attributions against next-word prediction (NWP) attributions across two naturalistic fMRI datasets, five pretrained models spanning transformers/SSMs/hybrids, and multiple attribution methods. The main claim is that BA and NWP rely on systematically different subsets of words: NWP emphasizes local/syntactic cues with sharp recency and primacy biases, while BA relies more on semantic/discourse information and broader recency integration.

**Compliance With Llm Reviewing Policy:**

Affirmed.

**Final Justification:**

The paper presents a useful attribution pipeline for brain–LLM alignment with broad empirical coverage, and the rebuttal partially addressed my concerns. However, the BA masking validation remains only partially convincing with respect to attribution selectivity, and some appendix results show more variability than the main narrative acknowledges, so I maintain my overall score of 4 while increasing my Soundness score to 3.

**Key Questions For Authors:**

Q1. Can the authors better justify that the observed attribution differences reflect differences between BA and NWP themselves, rather than differences in the downstream pipelines? In particular, how much of the effect could be driven by BA-specific components such as TR averaging, four-TR delay concatenation, the linear/ridge readout, and the voxelwise MSE objective, versus the LM head and next-token cross-entropy used for NWP?

Q2. Given that the empirical study is limited to two English narrative reading fMRI datasets and relatively small frozen LLMs, how broadly do the authors believe their conclusions should generalize? Relatedly, since higher-order semantic and discourse capabilities often shift with model scale, how do the authors expect the reported attribution differences to behave in larger models (e.g., 8B+)? Finally, several appendix results appear more mixed than the main narrative suggests, with BA sometimes showing recency or primacy peaks comparable to, or stronger than, NWP under certain attribution methods, architectures, or datasets; can the authors reconcile these cases more explicitly and clarify which aspects of the trend are method- or dataset-dependent?

Q3. Can the authors provide stronger evidence that the BA attribution maps identify uniquely important words, rather than reflecting the general fragility of the encoding setup to contextual disruption? In particular, since BA is already substantially degraded by random masking and same-POS masking of top-attributed words is often only modestly worse than the random baseline, how should readers interpret the current masking results in terms of attribution selectivity and faithfulness? It would be helpful if the authors could clarify what these experiments do and do not establish, and ideally strengthen this part with more direct faithfulness tests.

Q4. How do the authors reconcile their conclusion that BA favors semantic/discourse information with prior work such as AlKhamissi et al. (EMNLP 2025) and Mahowald et al. (2024, TiCS), which emphasize formal or syntactic competence as central to brain alignment? Can the paper more clearly explain whether the difference comes from methodology, dataset, operationalization of “importance,” or something else?

**Limitations:**

The paper already includes a reasonable limitations and impact discussion. I do not see a major missing discussion of negative societal impact.

**Strengths And Weaknesses:**

## Strengths

S1. The proposed attribution-based framing is reasonably novel for this setting.

S2. The empirical study is broad: two datasets, five models from multiple architecture families, multiple attribution methods, multiple layers, and several complementary analyses (IoU, spread, positional effects, feature categories, masking).


## Major Weaknesses

W1. Although both tasks use the same raw input context, BA and NWP are evaluated through substantially different computational pipelines. BA relies on TR-averaged final-word embeddings, concatenation across four delayed TRs to model the BOLD response, a linear/ridge encoding head, and an MSE loss over voxels, whereas NWP uses the LM head with a cross-entropy objective on the next word. As a result, the attribution differences may partly reflect differences in readout, temporal aggregation, and supervision signal, rather than purely differences between brain alignment and next-word prediction. In particular, the paper does not sufficiently discuss whether the temporal smoothing inherent to fMRI (and the explicit concatenation of four delayed TRs) could mechanically broaden BA attributions, thereby contributing to the higher attribution spread and broader recency profile relative to the more instantaneous NWP setup. The comparison may also be biased by the fact that representative layers are selected based on BA performance.

W2. The empirical scope is still fairly narrow: the study uses only two English narrative reading fMRI datasets, relatively small frozen language models, and coarse linguistic feature annotations available only for the Harry Potter dataset. Accordingly, claims about cross-architecture consistency or about the general relationship between BA and NWP should be stated more cautiously. The paper identifies an interesting pattern in this particular setting, but not yet a broadly established principle. This concern is especially relevant because the model set is limited to small LLMs (roughly 1B-1.5B parameters). Since higher-order semantic and discourse capabilities often change substantially with scale, it remains unclear whether the reported BA-NWP attribution differences would persist, widen, or diminish in larger contemporary models (e.g., 8B or 70B+). Moreover, some appendix figures appear less consistent with the main narrative: in several cases, BA seems to exhibit recency or primacy peaks as strong as, or stronger than, NWP using different architectures, under alternative attribution methods or on the Moth Radio Hour dataset (e.g., Figures 13, 14, 20, 21, 22), while in other plots the overall trend is difficult to interpret. This variability further suggests that the headline conclusions should be framed with more caution.

W3. The attribution validation is helpful, but not yet fully persuasive for BA. While the masking experiments show that perturbing highly attributed words degrades performance, BA also appears quite sensitive to random masking, and same-POS masking of top-attributed words is in several cases only slightly worse than the random baseline. As a result, the current evidence does not clearly establish that the method identifies uniquely important words for BA rather than reflecting the broader fragility of the encoding setup to contextual disruption.


## Minor Weaknesses

M1. The claim that BA favors semantic over syntactic information should be better situated relative to prior work. In particular, AlKhamissi et al. (EMNLP 2025) and Mahowald et al. (2024, TiCS) argue that formal competence, including syntactic relevance, may be more central to brain alignment than semantic or functional competence. Because these works use different methods, the disagreement may reflect differences in what is being probed rather than a direct contradiction. Still, the paper should discuss this tension explicitly and explain why its attribution-based framework yields a different conclusion.

M2. A disproportionate amount of text and analysis is dedicated to examining the "oscillatory" attribution pattern of Llama3.2-1B. Because the authors eventually conclude this is largely a stimulus-dependent gradient artifact, this section distracts from the core, generalizable findings of the paper and could be significantly condensed.

---

> ### Author Rebuttal · Authors · 2026-03-30
>
> _We thank the reviewer for the careful observations and constructive feedback._
>
> # Relation between BA/NWP attribution pipelines and attribution differences
> Because this concern overlaps with reviewer CwJs, we summarize the main argument here and provide the full clarification there. Briefly, our evidence argues against pipeline asymmetry as the primary driver because differences in attribution spread reverse across layer depths and selectively affect semantic/discourse features rather than the more frequent and broadly distributed syntactic category.
>
> # BA-based layer selection
> Representative layers were selected on Harry Potter and applied unchanged to Moth Radio Hour. If BA-based layer selection introduced a systematic bias, results should weaken on the second dataset. Instead they replicate closely, arguing against this confound.
>
> # Clarifications on method- or dataset-specific claims
> Our main positional findings are:
>
> 1. NWP consistently shows a sharp bimodal distribution;
> 2. BA exhibits a broader recency bias, reflected in a lower CoM;
> 3. Primacy effects are generally more pronounced for NWP.
>
> These trends hold across architectures on both datasets with Gradient X Input (Fig. 5, 22, 13, 14). NWP consistently shows a bimodal pattern across models. The recency peak is largely shared between tasks (visible in the intersection set), while the primacy peak is primarily driven by words uniquely important for NWP. BA shows a broader and more pronounced recency profile and a lower CoM. BA primacy appears only at high attribution thresholds (e.g., 80% in Fig. 14) and remains weaker overall.
>
> With alternative attribution methods, some architecture-specific deviations appear:
> - Falcon3 shows a clearer primacy peak for BA (Fig. 20–21), although BA still maintains a stronger recency bias overall (lower CoM than NWP).
> - Gemma and Zamba with SmoothGrad exhibit a broader BA distribution in which primacy becomes more prominent than for NWP.
>
> We will clarify that primacy magnitude is partly method- and architecture-dependent in the revised manuscript.
>
> # Generalizability to non-narrative datasets
> We agree our conclusions should be interpreted within currently available naturalistic datasets and tested model scales, and we will revise the manuscript accordingly.
>
> Long-form naturalistic narratives remain the dominant paradigm in brain-LLM alignment research because they maintain participant engagement during extended recordings. Although our experiments rely on two narrative datasets, they differ substantially: Harry Potter contains fictional literary text, while Moth Radio Hour consists of autobiographical spoken stories with different discourse structure and lexical statistics. The consistency across both suggests the effects are not stimulus-specific.
>
> # Generalizability to more diverse models
> During the review period, we extended the analysis to additional Gemma variants (2B-Instruct and 7B) and observed the same attribution patterns. These initial results suggest that the reported BA-NWP differences may extend beyond the 1-2B scale and to other training objectives. Full details are provided in the response to reviewer qxCS.
>
> # Masking results and BA attribution selectivity
> BA sensitivity to random masking is consistent with our finding that BA relies on distributed contextual evidence rather than a sparse set of critical words. Systems integrating information broadly across context degrade under both targeted and random masking. The fact that top-attributed masking consistently matches or exceeds random masking degradation proves the attribution pipeline identifies relative contribution of words to BA predictions, rather than sparse critical words for BA, which is sufficient for the comparative analyses we conduct.
>
> We clarify this interpretation in the revision.
>
> # Relationship to prior work
> Prior work shows that formal linguistic competence strongly predicts brain alignment across models. Importantly, word meaning, which underlies our semantic and discourse annotations, is itself part of formal competence in that framework (see Fig. 1 in Mahowald et al. (2024, TiCS)). Note that the discourse annotations we use identify individual words that allow for story progression, i.e. character names, action verbs, speech verbs, etc. By contrast, functional competence refers to broader abilities such as pragmatic reasoning, discourse coherence, and real-world language use, which are substantially more complex and cannot be assessed with our word-level attribution approach.
>
> Our analysis therefore does not test formal versus functional competence directly. Instead, it examines which contextual information within frozen representations most supports brain prediction. Additionally, our results show that semantic and discourse features are much more important for BA than NWP, but syntax still matters for BA (all red bars in Fig. 4 are always comparable). We will add this discussion to the paper.

---

> > ### Author Rebuttal · Reviewer_dw3b · 2026-04-03
> >
> > Thank you for the rebuttal. My concerns are partially resolved, but I still have two follow-up questions. First, while I agree that using lagged TRs is standard in BA, my concern is whether the BA-vs-NWP attribution comparison is robust to that design choice. Since the main claim of the paper depends on differences between BA and NWP attribution profiles, I think the paper would be stronger if it included a small robustness check showing that the main trends do not materially depend on the number of lagged TRs used in the BA pipeline.
> >
> > More specifically, could the authors provide a small ablation over the number of lagged TRs (e.g., 1 / 2 / 4), even on a subset of models or on one dataset, to show that the main attribution trends are not substantially driven by the BA temporal aggregation pipeline? I appreciate the rebuttal’s argument that four-TR temporal smoothing/concatenation is unlikely to be the main driver, but I still think a direct ablation would make this point much more convincing.
> >
> > Second, regarding Fig. 26(b), could the authors clarify exactly how the noise ceiling was computed and how normalized correlations were obtained? Recent methodological work suggests that split-half / Spearman-Brown reliability bounds explainable variance, whereas the appropriate ceiling for correlation is its square root. This would help interpret the values above 1.
> >
> > **[R1]** *How Much Variance Does Your Model Explain? A Clarifying Note On The Use Of Split-Half Reliability For Computing Noise Ceilings*, Sander van Bree, Malin Styrnal, Martin N. Hebart, 2025

---

> > > ### Author Response · Authors · 2026-04-05
> > >
> > > _We thank the reviewer for the follow-up questions and useful suggestions._
> > >
> > > ## Ablation on the number of lagged TRs
> > > After the reviewer's suggestion to empirically investigate whether the number of concatenated TRs in the BA estimation affects the attribution results,  we now have results for two models (Falcon3, Gemma-2B) on the Harry Potter dataset by concatenating 1 and 2 TRs, which we share at [this link](https://anonymous.4open.science/r/new_results-4819/README.md) under "trs_ablation”. The experiments for the other models are running and will be included in the revised manuscript.
> > >
> > > Results show that the core findings hold when reducing the number of concatenated TRs to both 1 and 2. Specifically:
> > > 1. Very low IoU at stringent attribution thresholds ($\leq 10$), but significantly above baseline IoU.
> > > 2. Opposing trends in attribution spread, with NWP AUC being significantly higher at early layers and BA AUC becoming significantly higher at late layers.
> > > 3. NWP relying more heavily on syntactic features, while BA draws equally from discourse and semantic contextual information.
> > > 4. Positional patterns replicate:
> > > - NWP consistently shows a sharp bimodal distribution (recency peak reflected in the Intersection set).
> > > - BA exhibits a broader recency bias, reflected in a lower CoM.
> > > - Primacy effects are generally more pronounced for NWP.
> > >
> > > ## Noise ceiling
> > > We have computed the noise ceiling following the procedure introduced in [1], commonly used on the Harry Potter dataset [e.g. 2,3,4] and more generally on naturalistic brain data. Split-half / Spearman–Brown reliability ceilings cannot be estimated as the dataset does not contain repeated trials for the same stimuli. According to [1], we estimate the ceiling by predicting the fMRI activity of a target participant from the data from other participants using linear models. This produces an estimate of the amount of stimulus-driven variance shared across participants for each voxel, rather than a within-subject reliability estimate. This definition differs from split-half / Spearman–Brown reliability ceilings, which estimate within-subject response reproducibility across repeated presentations of the same stimulus.
> > >
> > > Normalized correlations in Fig. 26(b) were obtained by dividing voxelwise prediction correlations by the corresponding voxelwise ceiling values (after excluding noisy voxels with ceiling < 0.05), as done in previous work [2,3,4]. Because the ceiling itself is an estimate derived from finite inter-subject data rather than a true upper bound, normalized values slightly above 1 can occur. Similar effects have been reported in prior work using the same ceiling estimation procedure on this dataset [3], highlighting the need for improved ceiling estimation and larger neural datasets.
> > >
> > >
> > > [1] Schrimpf, M., Blank, I. A., Tuckute, G., Kauf, C., Hosseini, E. A., Kanwisher, N., Tenenbaum, J. B., and Fedorenko, E. The neural architecture of language: Integrative modeling converges on predictive processing. Proceedings of the National Academy of Sciences, 118(45):e2105646118, 2021. doi: 10.1073/pnas.2105646118.
> > >
> > > [2] Aw, K. L. and Toneva, M. Training language models to summarize narratives improves brain alignment. ICLR, 2023.
> > >
> > > [3] Aw, K. L., Montariol, S., AlKhamissi, B., Schrimpf, M., & Bosselut, A. Instruction-tuning Aligns LLMs to the Human Brain. In First Conference on Language Modeling. 2023.
> > >
> > > [4] Merlin, G., Toneva, M. When Language Models Lose Their Mind: The Consequences of Brain Misalignment. ICLR, 2026.

---

### Decision · Program_Chairs · 2026-04-30

**Decision:**

Accept (regular)

**Comment:**

The paper puts forward  an approach for analyzing  brain–LLM alignment  by comparing brain alignment (BA) and next-work prediction (NWP), showing that the two rely on different subsets of input words, with brain alignment emphasizing semantic and discourse information. The work is solid, well-executed. Some concerns remain about limited novelty, potential confounds in comparisons,  a restricted dataset and model scale, and partially unconvincing validation of attribution faithfulness. The contribution is solid, even if limited in scope.